# Meta-Analysis of Public RNA Sequencing Data Revealed Potential Key Genes Associated with Reproductive Division of Labor in Social Hymenoptera and Termites

**DOI:** 10.3390/ijms24098353

**Published:** 2023-05-06

**Authors:** Kouhei Toga, Hidemasa Bono

**Affiliations:** 1Laboratory of BioDX, PtBio Co-Creation Research Center, Genome Editing Innovation Center, Hiroshima University, 3-10-23 Kagamiyama, Higashi-Hiroshima City 739-0046, Hiroshima, Japan; togchemi@hiroshima-u.ac.jp; 2Laboratory of Genome Informatics, Graduate School of Integrated Sciences for Life, Hiroshima University, 3-10-23 Kagamiyama, Higashi-Hiroshima City 739-0046, Hiroshima, Japan

**Keywords:** division of labor, eusociality, social insect, meta-analysis, RNA sequencing

## Abstract

Eusociality in insects has evolved independently many times. One of the most notable characteristics of eusociality is the reproductive division of labor. In social insects, the reproductive division of labor is accomplished by queens and workers. Transcriptome analyses of queens and workers have been conducted for various eusocial species. However, the genes that regulate the reproductive division of labor across all or multiple eusocial species have not yet been fully elucidated. Therefore, we conducted a meta-analysis using publicly available RNA-sequencing data from four major groups of social insects. In this meta-analysis, we collected 258 pairs (queen vs. worker) of RNA-sequencing data from 34 eusocial species. The meta-analysis identified a total of 20 genes that were differentially expressed in queens or workers. Out of these, 12 genes have not previously been reported to be involved in the reproductive division of labor. Functional annotation of these 20 genes in other organisms revealed that they could be regulators of behaviors and physiological states related to the reproductive division of labor. These 20 genes, revealed using massive datasets of numerous eusocial insects, may be key regulators of the reproductive division of labor.

## 1. Introduction

Organisms have undergone several evolutionary transitions from cellular to multicellular organisms and their organized societies [1]. Sociogenomics aims to comprehensively understand social life from a molecular perspective [2]. Eusociality is the most advanced social form, characterized by a high level of cooperation among members of society. An essential characteristic of eusociality is the reproductive division of labor, which is accomplished by reproductive and non-reproductive castes. Reproductive castes are physiologically and behaviorally specialized in reproduction, whereas non-reproductive castes engage in brood care or foraging. The division of labor is believed to increase the overall productivity of eusocial colonies [3].

Eusociality, which is based on the reproductive division of labor, has evolved independently in several insect lineages [4,5]. Eusocial Hymenoptera (bees, ants, and wasps) and termites are extensively studied insects that have complex and sophisticated societies. The reproductive and non-reproductive castes of these social insects are referred to as queen/king and workers/soldiers, respectively [6]. The investigation of genomic changes underlying the convergent evolution of eusociality among several lineages (bees and ants) has identified the genes in which molecular evolution has occurred with increased social complexity [7,8,9,10]. Transcriptome analysis using the pharaoh ant *Monomorium pharaonis* and the honeybee *Apis mellifera* revealed that convergent eusocial evolution is based on the expression of highly conserved reproductive and lineage-specific genes [11]. In addition, many transcriptome analyses of queens and workers have been conducted for each species of bee, ant, wasp, and termite [12,13,14,15,16,17,18,19,20,21,22,23,24,25,26,27,28,29,30,31,32,33]. However, no comprehensive transcriptome analysis has been performed to understand the regulatory mechanisms of reproductive division of labor across a wide range of eusocial species.

Physiological differences between queens and workers are mainly produced by quantitative differences in juvenile hormones (JH), neurotransmitters, and neuropeptides. JH generally acts as a gonadotropic hormone in eusocial and solitary insects [34,35,36,37,38]. Vitellogenin (Vg) is precursor protein of the egg yolk and is taken up by oocytes via receptor (Vg receptor)-mediated endocytosis [39,40]. Vg synthesis and uptake into ovary are promoted by JH [41]. Vg and its receptors are highly expressed in the reproductive castes of various social insects [11,30,42,43,44,45,46,47]. The regulation of the secretion of neuropeptide and neurotransmitter plays a central role in controlling the reproductive division of labor. The expression of insulin-like peptide (ILP) is upregulated in the queens of several ants and termite *Macrotermes natalensis* [48,49]. In *A. mellifera*, however, *ILP* expression is lower in the old queen than in the old workers [50]. Neuropeptides other than insulin also play an important role in the reproductive division of labor. Corazonin is a neuropeptide that is highly expressed in the workers of three ants (*Harpegnathos saltator*, *Camponotus floridanus*, and *Monomorium pharaonis*) and one wasp (*Polistes canadensis*) [16]. Biogenic amines are the primary neuroactive substances that control the ovary development of reproductive caste and worker behavior [51].

A meta-analysis is common in clinical research but less general in basic biology including entomology [52]. Recently, however, a meta-analysis has provided interesting insights into several fields of entomology. A meta-analysis of the number of insect species provided a different view of spatiotemporal patterns of insect abundance than previously suggested. In the field of genomics, a search for horizontally transmitted genes within many numbers of insect genomes led to the discovery of genes involved in courtship behavior in lepidopteran insects. A meta-analysis that combines transcriptomic data obtained from multiple studies is effective in identifying novel gene expression associated with specific biological processes [53,54,55]. Additionally, it is possible to identify master regulatory genes by combining transcriptome data from a wide range of species. We developed meta-analysis methods for publicly available transcriptome data from humans, plants, and insects [53,56,57,58,59]. In this study, we used a meta-analysis method to identify regulatory genes related to the reproductive division of labor in ants, bees, wasps, and termites. We collected RNA sequencing data for 258 pairs (queens vs. workers) from 34 eusocial species. The meta-analysis detected 212 genes that were conserved in all species examined in this study; 20 of these 212 genes were significantly differentially expressed between queens and workers.

## 2. Results

### 2.1. Data Collection of Transcriptomes Related to Queens and Workers

RNA sequencing data of queens and workers have been collected from many social insects to identify the genes underlying the reproductive division of labor. Consequently, 258 pairs (queen vs. worker) of RNA sequencing data were collected from 34 eusocial insect species (Figure 1A, Appendix A; https://doi.org/10.6084/m9.figshare.21524286.v6, accessed on 6 April 2023). Ants (*Harpegnathos saltator* and *Monomorium pharaonis*), termites (*Cryptotermes secundus*), honeybees (*Apis. mellifera*), and wasps (*Polistes dominula*) were among the higher ranked species, indicating that RNA sequencing data from a wide range of eusocial species were included in the dataset. When compared among the tissues, brain-derived RNA sequencing data were the most abundant (Figure 1B).

### 2.2. Meta-Analysis of RNA Sequencing Data in Social Insects

The expression ratio (queen vs. worker) (QW ratio) was calculated for each species. At this point, the transcript IDs in the QW ratio table differ from species to species, even for homologs. Therefore, we converted the transcript IDs of each species to the protein IDs of *A. mellifera*, and then merged the tables in which the QW ratio of each species was described. Among the social insects used in this study, 212 genes were conserved. In the process of confirming homology, transcripts that were not translated into proteins were found but were not included in this study (Appendix A: https://doi.org/10.6084/m9.figshare.21524286.v6, accessed on 6 April 2023). In addition, for each of 212 genes, 258 pairs of QW ratio values were obtained (Appendix A: https://doi.org/10.6084/m9.figshare.21524286.v6, accessed on 6 April 2023). Changes in the number of genes commonly present among species are shown in Appendix A. The QW score for each gene was then calculated based on the QW ratio table. The QW score for each gene was calculated by subtracting the number of pairs with downregulated (less than 1/1.5-fold change) gene expression from the number of pairs with upregulated (more than 1.5-fold change) gene expression. This method provides the overall trends of gene expression changes in 258 pairs of RNA sequencing data. High scores indicate highly expressed genes in the queen, while low scores indicate highly expressed genes in the workers. The 212 identified genes were ranked based on their QW score values (Appendix A: https://doi.org/10.6084/m9.figshare.21524286.v6, accessed on 6 April 2023). These genes were annotated based on the corresponding homologs of other species (*D. melanogaster*, *H. sapiens*, *Mus musculus*, *Bombyx mori*, *Bombus terrestis*, and *Nasonia vitripennis*) using the annotation table created by Yokoi et al. 2022 [60] (Appendix A: https://doi.org/10.6084/m9.figshare.21524286.v6, accessed on 6 April 2023). We focused on the top 10 and bottom 10 genes in Appendix A because they showed abrupt score changes (Figure 2 and Table 1). Abrupt score changes indicated that their expression was remarkably variable in the dataset used in this study. The QW scores for each species are available in Appendix A (https://doi.org/10.6084/m9.figshare.21524286.v6, accessed on 6 April 2023). The QW score of *H. saltor* is not extremely high or low compared with other species, despite this species have many replicates (Figure 1). Therefore, it was unlikely that the QW score was biased by the number of replicates of *H. saltor.*

In this study, the number of genes evaluated was small (212 genes). To show that the top 10 and bottom 10 genes remained important even if the number of genes evaluated changes, their ranking was investigated when the number of genes to be evaluated was increased. Based on the information from Appendix A, we excluded three expression data (*Temnothorax longispinosus2, Reticulitermes speratus,* and *Macrotemes natalensis*) to prevent an extreme reduction in the number of genes to be evaluated. As a result, the number of genes evaluated was increased to 2595. Within the gene set, the top 10 and bottom 10 genes ranked within the top 6% and bottom 8%, respectively (Appendix A). The plots showing the ranking of the top 10 and bottom 10 genes were located on the periphery or outside where the scatter plot changed abruptly (Appendix A). These results show that an increase in the number of genes evaluated did not drastically reduce the ranking of these genes.

To estimate the function of these genes, information on the tissue-expression profile in *D. melanogaster* was added (Table 1). Nine out of ten highly expressed genes in workers were found primarily in the brain and thoracoabdominal ganglia in *D. melanogaster,* while highly expressed genes in queens were found in various tissues. If a high expression had already been reported in queens or workers, caste and species names were noted. In addition, we added the biological/molecular functions predicted by the cited literature.

### 2.3. Enrichment Analysis

We selected the top 30 best-ranked and worst-ranked genes based on the QW score, and then performed an enrichment analysis using the corresponding IDs of *D. melanogaster* and *H. sapiens* (Figure 3). The molecular functions of many genes are highly conserved across species. In addition, since there is abundant information on human gene function, we can use this to infer the gene function of social insects. The number 30 was set arbitrarily. This setting was determined with reference to the scatterplot curve shown in Figure 2. No GO terms with markedly low *p*-values (−log10(P) values greater than 4) were detected when *D. melanogaster* IDs were used (Figure 3A,B). However, when *H. sapiens* IDs were used, burn wound healing was significantly enriched in the top 30 genes (Figure 3C), whereas the positive regulation of transmembrane transport and modulation of chemical synaptic transmission were enriched in the last 30 genes (Figure 3D).

## 3. Discussion

Variation in gene expression can be caused by a variety of factors, including natural variation and measurement methods. This causes a reduction in statistical power, leading to overlooked differentially expressed genes. In general, effective strategies to address this problem include increasing sample size and analyzing data in a uniform manner. Meta-analyses combine the results of multiple studies into a single analysis, thereby increasing sample size and statistical power to detect differentially expressed genes. In this study, the meta-analysis allowed us to identify several differentially expressed genes that have not been reported, as described below.

In this study, we aimed to identify genes underlying the reproductive division of labor across all or multiple eusocial insects by performing a meta-analysis of RNA-seq data. The meta-analysis revealed 20 genes with differential expression between queens and workers. Among these genes, vitellogenin and vitellogenin receptors, which are highly expressed in queens across many social insects [11,30,42,43,44,45,46,47], showed the highest QW score (Figure 1). This suggests that the current meta-analysis method is suitable for identifying differences in gene expression between queens and workers.

Among the genes identified in this study, we focused on 20 genes with the highest or lowest QW scores (Table 1). As the results of this study represent a vast amount of data from a wide variety of eusocial species, it is likely that these genes are the key factors underlying the reproductive division of labor in many social insects. In addition, several of the 20 genes were not previously implicated in the reproductive division of labor, providing novel insights into the underlying mechanisms of the reproductive division of labor. In the following sections, we summarize the biological and molecular functions of the corresponding homologs of these 20 genes in other organisms and discuss their role in social insects. Topics other than those listed below were available in the Appendix A (Supplemental Discussion DOI: https://doi.org/10.6084/m9.figshare.21524286.v6, accessed on 6 April 2023).

### 3.1. Upregulated Genes in Queen

It has been known that the reproductive performance of a queen is mainly regulated by JH (+ vitellogenin) and insulin in several social insects. We found several genes involved in these processes, as discussed below.

#### 3.1.1. Oogenesis, Juvenile Hormone (JH) Binding, and Synthesis (*Vitellogenin*, *yolkless*, *Cyp305a1*, *apolipophorin*, and *exuperantia*)

In queens, the expression of JH-related genes (*Vitellogenin, yolkless, Cyp305a1*, and *apolipophorin*) and oogenesis gene (*exuperantia*) was upregulated as expected. *Cyp305a1*, which encodes JH epoxidase, is required for JH biosynthesis in *D. melanogaster* [66]. *apolipophorin* is the gene encoding a JH-binding protein in the cockroach *Blattella germanica* [61], and is upregulated in the termite queens, *Reticuliterms speratus* [31]. *exuperantia* is a maternal factor that is involved in oocyte formation [63]. These genes may contribute to queen ovarian development via an upregulated JH titer. Although these genes, except for *Vitellogenin* and *Vitellogenin receptor* (*yolkless*), are known to be expressed in the small number of species (Table 1), this study suggests that these genes are involved in the queen ovary development of many social insects. However, in the honeybees *A. mellifera* and the ant *H. saltator,* the gonadotropic function of JH was lost [80,81]. The gonadotropic effects of these genes may depend on the species.

#### 3.1.2. The Regulation of Insulin Secretion (SPARC (Secreted Protein Acidic and Cysteine Rich) and RSG7 (Regulator of G Protein Signaling 7))

The expression of insulin-like peptide (ILP) is upregulated in the queens of several ants and in the termite *Macrotermes natalensis* [48,49]. To date, however, the regulatory mechanisms of insulin secretion in social insects are largely unknown. In this study, *SPARC* was highly expressed in queens, and *RSG7* was highly expressed in workers. In mice, *SPARC* promotes insulin secretion via the downregulation of *regulator of G protein 4* (*RGS4*) expression in pancreatic β cells [64]. We speculate that *SPARC* may also regulate insulin secretion in eusocial insects via the downregulation of *RSG7* expression, but this requires further study. The discovery of *SPARC* and *RSG7* is a major achievement of the meta-analysis because these genes have not been reported as regulators of insulin secretion in social insects. In *A. mellifera*, however, *ILP* expression is lower in the old queen than in the old workers [50]. Therefore, the relationship between *SPARC* and *RGS7* may possibly not apply to all eusocial species.

An enrichment analysis showed that burn wound healing (WP5055) was enriched in the queen, and *SPARC* was part of this pathway. Because the relationship between this pathway and the queen trait is unknown, functional analyses are expected.

### 3.2. Upregulated Genes in Workers

Tissue expression profiles in *D. melanogaster* were referred for the 20 listed genes using FlyAtlas2 [82] (accessed on 14 October 2022). Unlike the genes significantly expressed in queens, many of the gene sets upregulated in workers were expressed in the nervous system of adult *D. melanogaster* (Table 1). The enrichment analysis also suggests the importance of genes expressed in the nervous system. Genes involved in the positive regulation of transmembrane transport (GO:0034764) were enriched in the results of the enrichment analysis. It has been reported that genes related to membrane and ion transport are also expressed in the brain of *Polistes canadensis* [83], suggesting the contribution to increased cell activity in the brain. In addition, the enrichment analysis showed that genes involved in the modulation of chemical synaptic transmission (GO:0050804) were enriched in the gene set that is highly expressed in workers. On the other hand, genes expressed in the nervous system were rarely present among the genes highly expressed in queens (Table 1 and Figure 3). These results suggest active gene expression in the nervous system of workers, leading to worker-specific behaviors. In the following sections, we discuss the genes that may be associated with worker-specific behaviors.

#### 3.2.1. The Regulation of Secretion of Neuropeptide and Neurotransmitter (*amon* (Amontillado) and *Syt1* (Synaptotagmin 1))

*amon* may be involved in the secretion of neuropeptide related to determining worker behaviors. *amon*, which encodes a member of the prohormone convertase family, is required for propeptide processing to form bioactive neuropeptides [77]. In *D. melanogaster*, the amon protein is an important enzyme that catalyzes many neuropeptides in the larval corpora cardiaca and perineuronal organs of the thorax and abdomen. Corazonin is a neuropeptide whose production is dependent on the catalysis of the amon protein [77]. Actually, corazonin plays a central role in determining the worker identity in *Harpegnathos saltator* and is also highly expressed in the workers of three ants (*Harpegnathos saltator*, *Camponotus floridanus*, and *Monomorium pharaonis*) and one wasp (*Polistes canadensis*) [16]. In addition to corazonin, *amon* may regulate the production of neuropeptides that regulate worker behavior. This gene has not been reported to control worker behavior in social insects.

Serotonin secretion regulates worker responsiveness to trail pheromones in the ant *Pheidole dentata*, leading to cooperative foraging behavior [84]. *Syt1* may control the secretion of neurotransmitters such as serotonin involved in the regulation of social behavior. *Syt1,* which encodes a synaptic vesicle calcium-binding protein, is required for synaptic vesicles to interact with the plasma membrane [71]. Ca2^+^ ions act on molecular complexes, including *Syt1*, triggering rapid exocytosis at the synapse. *Syt1* is highly expressed in ants (*S. invicta*), honeybees (*A. mellifera*), and wasps (*P. metricus*) [70]. *Syt1* might regulate the secretion of neurotransmitters that cause foraging and brood care.

#### 3.2.2. Regulation of Circadian Sleep (*Gat* (*GABA Transporter*), *quiver* (*qvr*), and *CG30158*)

In honeybees, *A. mellifera*, sleep deprivation negatively affects the accuracy of the waggle dance, which transmits the location of food resources to colony members [85]. This suggests that sleep control is essential for organized social behavior. *qvr,* which encodes a glycosylphosphatidylinositol (GPI)-anchored membrane protein, is a sleep-promoting factor in *D. melanogaster* [75]. *Gat* (*GABA transporter*) encodes a protein belonging to the solute carrier transporter family. Mutations in *Gat* cause excessive sleep because of the reduced uptake of GABA (a sleep inhibitor) from synapses to astrocytes in *D. melanogaster* [76]. It is possible that the amount of sleep of workers was optimized by *Gat* and *qvr*. Although these two genes are known to be expressed in workers of two bee species [62,74], the results of this study suggest that they may be involved in worker sleep regulation in many social insect species. It is unknown what effect excessive sleep has on the task performance of workers, but it is believed that the longer workers sleep, the less time they spend foraging and providing brood care for the colony.

A cooperative division of labor within the colony should require unifying the timing of when workers engage in foraging and feeding. *CG30158* is homologous to mammalian *Dexras1*, which encodes a Ras-like G protein. *Dexras1* is required for entrainment of circadian rhythms in the environmental light cycle in mice [72,73]. In worker foragers of the honeybee, *A. cerana japonica*, the circadian rhythm of locomotion activity is synchronized with light and temperature [86]. The *A. mellifera* foragers are often inactive at night [87]. In addition, the sleep rhythms of *A. mellifera* foragers are also known to be synchronized with the timing of the food available [88]. In social insects, the homolog of *CG30158* may play an important role in synchronizing foraging timing with environmental cycles.

## 4. Materials and Methods

### 4.1. Curation of Public RNA Sequencing Data

Accession IDs (Project ID) related to eusociality were searched from the public databases All Of gene Expression (AOE) [89] and DBCLS SRA (https://sra.dbcls.jp/ (accessed on 11 May 2022)) using the keywords caste, eusocial, eusociality, and sociality. Metadata were obtained using NCBI SRA Run Selector (https://www.ncbi.nlm.nih.gov/Traces/study/ (accessed on 12 May 2022)). The acquisition of RNA sequencing data was limited to species for which the genome or reference transcriptome was available. RNA sequencing data from each species were obtained in pairs (queen and worker) to investigate gene expression related to the reproductive division of labor.

The names of the castes whose expressions were compared are listed in Appendix A. Individuals that exclusively reproduce within a colony were defined as queens. The designation of the queen caste varies according to stage and species. In this study, “dominant”, “Gynes”, and “Gamergate” were treated as queens. “Subordinate” is treated as workers. In the case of social hymenopterans, workers behave differently depending on their age (nurse or forager). Therefore, comparative analyses of gene expression (queen vs. worker) were performed considering age (e.g., mated queen vs. foragers and virgin queen vs. nurse) (Appendix A). In termites, secondary (or neotenic) queens were also included as queens. These queens differentiate from offspring after the death of the primary queen within their nest.

### 4.2. RNA-Seq Data Retrieval, Processing, and Quantification

SRA retrieval and conversion to FASTQ files were performed using the prefetch and fasterq-dump programs in the SRA Toolkit (v3.0.0) in the Docker environment (v3.3.1) [90]. To decrease the disk space usage, the obtained FASTQ files were immediately compressed using pigz (v. 2.4). For trimming and quality control, Trim Galore! (v.0.6.6) [91] and Cutadapt (v.3.4) [92] software were used. Trim Galore! Was run with the parameter fastqc–trim1–paired (if data were paired-end). Trim Galore! Included Cutadapt (v.3.4) to trim low-quality base calls. Because all the data passed through this quality check, low-quality data were not included in this study. Biological replicates were treated as one experiment, and technical replicates were combined into one expression dataset.

The files of the reference genome and transcripts used for quantifying gene expression levels are listed in Appendix A (https://doi.org/10.6084/m9.figshare.21524286.v6 (accessed on 12 May 2022)). If available, these data were obtained from the NCBI Reference Sequence Database (RefSeq) or open data repositories used in the cited literature (accessed on 5 August 2022). Transcript expression was quantified using Salmon (v.1.5.0) [93] with the parameters -i index -l IU. Gene expression levels were quantified using transcripts per million (TPM). For species with only genome sequences, RNA sequencing reads were mapped to the reference genome using HISAT2 (v.2.2.1) [94] with the parameters -q --dta -x, and then, quantification of the mapped data was performed using StringTie (2.1.5) [95]. TPM was used as a quantitative value of the gene expression level.

### 4.3. The Detection of Genes Related to the Reproductive Division of Labor

A calculation method developed for the meta-analysis of RNA sequencing data was employed [53,56,57,58,59] as described below. Pairs of the queen and worker RNA sequencing data (Appendix A: https://doi.org/10.6084/m9.figshare.21524286.v6, accessed on 6 April 2023) were used to calculate the queen/worker ratio of TPM for each transcript (referred to as the QW ratio). The QW ratio for each transcript was calculated using Equation (1).
(1)$$QW\;ratio={log}_{2}\left(TPM_{queen}+0.01\right)-{log}_{2}\left(TPM_{worker}+0.01\right),$$

This pairwise calculation was performed for all the pairs described in Appendix A (https://doi.org/10.6084/m9.figshare.21524286.v6, accessed on 6 April 2023). The value “0.01” was added to TPM to convert zero into a logarithm. QW ratios were classified as “upregulated,” “downregulated,” or “unchanged” according to the thresholds. Transcripts with a QW ratio greater than 0.58496250072116 (1.5-fold expression change) were treated as “upregulated.” Transcripts with QW ratios less than −0.58496250072116 (1/1.5-fold expression changes) were treated as “downregulated.” Others were treated as “unchanged.” The QW score was calculated to identify the genes involved in the reproductive division of labor. The QW score was calculated by subtracting the number of pairs with “downregulated” from those with “upregulated,” as shown in Equation (2):(2)$$QW\;score=count\;{number}_{upregulated}-count\;number_{downregulated},$$

Python codes for calculating QW ratios and QW scores were obtained from a previous study [56] (https://github.com/no85j/hypoxia_code/blob/master/CodingGene/HN-score.ipynb (accessed on 13 August 2022)).

### 4.4. Annotation for Transcripts and Enrichment Analysis

To determine homologous relationships, the protein BLAST (BLASTP) program in the NCBI BLAST software package (v2.6.0) was used with parameters -evalue 1 × 10^−10^ -outfmt 6 -max_target_seqs 1. The transcript files used for protein translation are listed in Appendix A (https://doi.org/10.6084/m9.figshare.21524286.v6). For species with only genome sequences, GFFread (v0.12.1) [96] was used to extract transcript sequences from the genome sequence. TransDecoder (v5.5.0) (https://transdecoder.github.io/ (accessed on 22 June 2021)) was used to identify and translate the coding regions into amino acid sequences. Open reading frames were extracted using TransDecoder.Longorfs with the parameter -m 100, and coding regions were predicted using TransDecoder.Predict with the default parameters. As a result of the above processes, the transcript IDs of all species were converted to the protein IDs of *A. mellifera*. The protein IDs of *A. mellifera* were then converted to those of other species using the annotation table for transcripts of *A. mellifera* [60]. Transcripts and amino acid sequences created in this study can be downloaded from figshare (Supplemental File S1). The enrichment analysis was performed using the Metascape software with default settings (GO Biological Processes, Canonical Pathways, KEGG Pathway and WikiPathways with *p*-value < 0.01, a minimum count of 3 and an enrichment factor >1.5 (the ratio between the observed counts and the counts expected by chance)) [97]. The gene symbols for *D. melanogaster* and *H. sapiens* were used for Metascape.

## 5. Conclusions

This study is the first meta-analysis conducted on massive datasets of numerous eusocial species in two phylogenetically distant lineages, social Hymenoptera and termites, and provides two novel findings. First, the meta-analysis found 12 genes that had not been reported to be associated with the reproductive division of labor. This is because a meta-analysis removes the data bias that occurs in the results of individual studies. Second, the 20 genes retrieved from massive datasets of a large number of eusocial species may be key regulators of the reproductive division of labor. Functional analyses of the genes identified in this study are expected in the future. As there is the increase in the publicly available RNA sequencing data, meta-analyses should prove useful in providing new insights into targeted biological phenomena. Although the fact that meta-analysis of RNA sequencing data is hardly widespread in entomology, the present study demonstrated the effectiveness of a meta-analysis of RNA sequencing data.

## Figures and Tables

**Figure 1 ijms-24-08353-f001:**
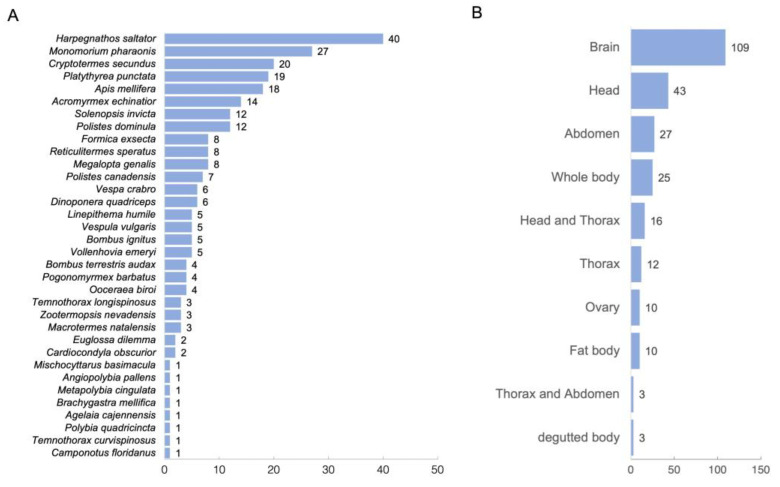
Number of RNA sequencing data compared across species (**A**) and tissues (**B**). The number of RNA-seq data used is indicated by bar graphs and numerical values.

**Figure 2 ijms-24-08353-f002:**
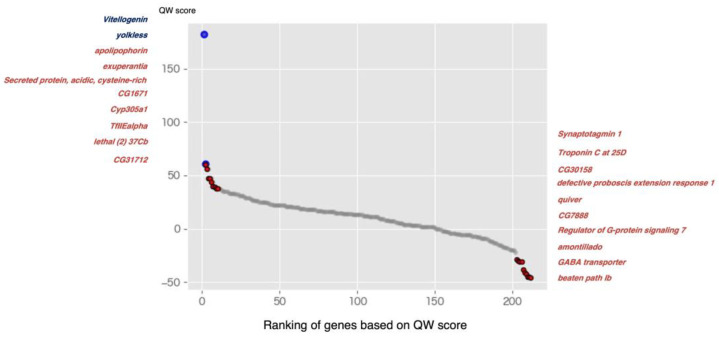
Scatter plots of QW score of all genes (212 genes) identified in this study. Gene ranking based on QW score values. The gene names follow those of *D. melanogaster*. Red and blue dots indicate the top 10 best-ranked and worst-ranked genes, respectively. The blue dots indicate *Vitellogenin* and *yolkless* (Vitellogenin receptor), which are significantly expressed in queens of many eusocial species. Gene names indicate the top 10 and bottom 10 genes.

**Figure 3 ijms-24-08353-f003:**
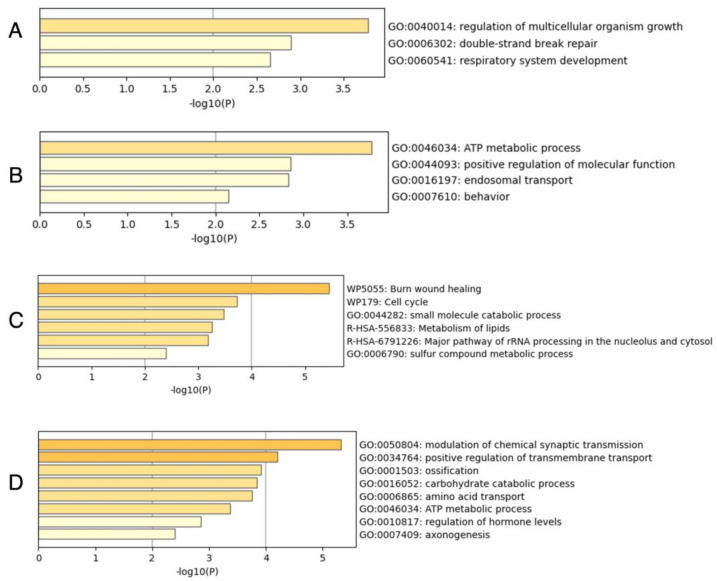
Results of enrichment analysis using *D. melanogaster* IDs (**A**,**B**) and *H. sapiens* IDs (**C**,**D**). Bars are colored differently according to the value of −log10(P). In this study, we focused on GO terms with −log10(P) values greater than 4.

**Table 1 ijms-24-08353-t001:** Top 10 best-ranked and worst-ranked genes based on QW score values.

RANK	QW Score	Gene ID of *A. mellifera*	Gene Symbol of *D. melanogaster*	Gene Symbol of *H. sapiens*	Tissue-Expression Profile in Female Adult *D. melanogaster*	Caste with High Expression that Was Reported in Previous Study	Biological/ Molecular Function
1	182	NP_001011578.1 (Vitellogenin)	-	-	-	Queen in many species [42,43,44,45,47]	Oogenesis [39,40]
2	61	XP_026295652.1	yl	LRP2	ovary	Queen in *Diacamma* sp.; *A. mellifera*; *M. pharaonis*; *T. longispinosus* [11,30,47]	Oogenesis [39,40]
3	60	XP_026298285.1	apolpp	GTPBP2	fat body, spermatheca, head, eye	Queen in *R. speratus* [31]	JH binding [61]
4	56	XP_623977.2	exu	-	ovary	Queen in *A. mellifera*; *T. longispinosus* [30,62]	Oogenesis [63]
5	47	XP_623079.2	SPARC	SPARC	heart, eye, head thoracicoabdominal ganglion	-	Insulin secretion [64]
6	47	XP_001120493.3	CG1671	TBL3	relatively high in ovary	-	Ribosome biogenesis [65]
7	44	NP_001314895.1	Cyp305a1	CYP2A13	spermatheca, head	-	JH synthesis [66]
8	40	XP_395253.2	TfIIEalpha	GTF2E1	relatively high in ovary	-	Ribosome biogenesis [67]
9	39	XP_026300350.1	lethal (2) 37Cb	DHX16	relatively high in heart	-	Splicesome [68]
10	38	XP_395437.4	CG31712	GKAP1	brain, thoracicoabdominal ganglion, head	-	Splicesome [69]
203	−29	XP_026295297.1	Syt1	SYT1	thoracicoabdominal ganglion, brain, head	worker (forager) in *A. mellifera*; *P. metricus*; *S. invicta* [70]	Neurotransmitter release at synapses [71]
204	−30	XP_026299328.1	TpnC25D	CALM3	head, eye, gut	-	A calcium signal transduction pathway (Uniprot)
205	−31	XP_026298569.1	CG30158	RASD1	brain, head, thoracicoabdominal ganglion	-	Circadian entrainment [72,73]
206	−31	XP_026300722.1	dpr1	CNTN4	brain, thoracicoabdominal ganglion, head	-	Immunoglobulin superfamily [74]
207	−38	XP_026302000.1	qvr	LYPD6	thoracicoabdominal ganglion, head, eye	worker (forager) in *B. terrestris* [75]	Sleep [76]
208	−41	XP_393138.2	CG7888	SLC36A1	hindgut, thoracicoabdominal ganglion, brain	-	Amino acid transport (Uniprot)
209	−42	XP_393403.3	RSG7	RGS7	brain, thoracicoabdominal ganglion, head	-	Insulin secretion [64]
210	−45	XP_006571520.1	Gat	SLC6A1	thoracicoabdominal ganglion, brain, head	Worker in *A. mellifera* [62]	regulation of circadian sleep [77]
211	−45	XP_392366.2	amon	PCSK2	brain, thoracicoabdominal ganglion, head	-	The production of bioactive neuropeptide hormones [78]
212	−46	XP_026298737.1	beat	PSG4	brain, thoracicoabdominal ganglion, head	-	Immunoglobulin superfamily [74,79]

## Data Availability

The data presented in this study are openly available in figshare (DOI: https://doi.org/10.6084/m9.figshare.21524286.v6 (accessed on 6 April 2023)).

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
