# Peer review of "Meta-Analysis of Public RNA Sequencing Data Revealed Potential Key Genes Associated with Reproductive Division of Labor in Social Hymenoptera and Termites"

_ijms, 2023, doi:10.3390/ijms24098353_

Round 1

Reviewer 1 Report (Previous Reviewer 3)

In the revised manuscript, authors have addressed all of my concerns by clarification. 

Reviewer 2 Report (New Reviewer)

The manuscript is a challenging approach to perform the meta-analysis using massive RNA-seq datasets. Although the research design have room for improvement and the conclusion is not so outstanding, this ambitious attempt is worth to be published in present form. 

Since the evolution of eusociality in the lineages to termites, ants, bees and wasps has evolved independently, the genetic transition processes from their solitary ancestor are different from each other. The authors concentrated on coincident QW-score behavior among orthologous genes, however, it is likely that genes responsible for this convergent evolution are occasionally selected from possible combinations.

I think that it is better to find conserved gene functions among genes selected by higher or lower QW-score threshold. As discussed in Bernadou A,et al. (Front. Ecol. Evol. 9:732907.), the key feature of social insects is the fecundity and longevity of queen. 

Furthermore, many of the examined species are ants, leading to the overweighing of this monophyletic lineage. I hope that the authors could refine the strategy of meta-analysis and narrow down the candidate genes.

This manuscript is a resubmission of an earlier submission. The following is a list of the peer review reports and author responses from that submission.

Round 1

Reviewer 1 Report

The authors conducted a meta-analysis using publicly available RNA-sequencing data from four major groups of social insects to identified genes differentially expressed in queens or workers in social Hymenoptera and termites. This topic is interesting. And the results showed total of 20 genes were differentially expressed in queens or workers. and 12 new genes were identified in the reproductive division of labor. However, only some RNA-sequencing data analyses were used in this research, and none molecular biology functional analysis to prove is the biggest imperfection. Overall, the contribution of the manuscript is very limit. I feel the paper has not reached the level of this to be considered for publication in this high-quality journal.

Author Response

We sincerely appreciate your evaluation. We agree with the reviewer that functional analysis of candidate genes is essential as pointed out by the reviewer, but we think that functional analysis is an issue for the future. The significance of finding candidate gene sets is evident from many recent reports of gene identification by RNA-seq analysis. The RNA-seq data we used is a large dataset of 258 pairs (queens and workers) from 34 species, not just some RNA-seq data. Meta-analysis using such large dataset offers the potential to discover genes that have been overlooked in individual studies before gene function analysis is performed. We believe that the discovery of such gene sets is a sufficient contribution to molecular biology.

Reviewer 2 Report

The title just showed a method used for an object of study. The results were too simple and is insufficient to support the conclusion. The paper cannot be accepted.

Author Response

Thank you very much for your evaluation. We have reconsidered the title as follows:

“Meta-analysis of public RNA sequencing data revealed potential key genes associated with reproductive division of labor in social Hymenoptera and termites”

The significance of finding candidate gene sets is evident from many recent reports of gene identification by RNA-seq analysis. The RNA-seq data we used is a large dataset of 258 pairs (queens and workers) from 34 species, not just some RNA-seq data. The gene set revealed by our meta-analysis using the large data set is sufficiently reliable. This sufficiently supports our conclusion that we have identified 20 candidate genes as the potential key factors associated with reproductive division of labor.

Reviewer 3 Report

This MS entitled “Meta-analysis of public RNA sequencing data of queens and workers in social Hymenoptera and termites” by Toga and Bono describes that meta-analysis of RNA sequences is useful for searching for genes which are involved in biological process to be analyzed. Using this method, they tried to identify the genes which are involved in caste differentiation of social insects. The authors found newly 12 genes which might be involved in reproductive division between queens and workers. Using this method, the authors also extracted the genes which have already reported to involvement in reproductive division. The MS demonstrated the usefulness of the method for analysis of phenomena conserved across species. However, there has not been sufficient discussion as to how the analysis by this method yielded different results compared to the usual comparison method.  

Major concern,  

this MS has a methodological aspect. It is better to discuss why newly 12 genes were obtained as candidates for reproductive division by comparing multiple species with a large amount of data, in contrast with conventional comparisons. Please discuss the factors/process that led to the new results focused on the differences in the process of data analysis.

Below are some minor points that need to be correct before the MS is accepted.

1. In figure 3 legend, line 165-166, “H. sapiens IDs (B, C)” is correct? 

2. Line 303 and 335, why URLs are shadow?

3. In the references section, some papers lack page numbers or published year.

Round 2

Reviewer 1 Report

The authors have made substantive changes and clarifications to this manuscript resulting in a substantially improved manuscript dealing with this topical subject of  interest. I feel that they have addressed my major points through either additional experimentation or clarification in the text. 

Author Response

We thank the reviewer for your evaluation.

Reviewer 2 Report

The author collected 258 pairs (queen vs. worker) of RNA sequencing data from different species and performed meta-analysis and enrichment analysis simply. There were some questions that hampers publication of the study. I listed below:

1. The results were too simple.

2. The authors should consider the key factors and key stages for the reproductive division when they collected data. And it will be better to analyze the switch from key larval stage to reproductive division stages.

3. There were 212 genes were conserved among the social insects in this study. How many genes were lost when they converted the transcript IDs of each species to the protein IDs?

4. The enrichment analysis of the top 30 best-ranked and worst-ranked genes based on the QW score didn’t lead to significant results to understand the question of reproductive division.

Reviewer 3 Report

In the revised manuscript, authors have addressed all of my concerns by clarification. But some shadowed sentences remained, which should be corrected prior to publication.

There are shadowed sentences in line 119, 123, 315, 347, 357, and 388-392.

Author Response

Thank you very much for your evaluation and pointing this out. We revised the manuscript.

Round 3

Reviewer 2 Report

I checked the data the authors showed in table S1. The data and analyses were insufficient to support the conclusion.